# High DDT resistance without apparent association to kdr and Glutathione-S-transferase (GST) gene mutations in *Aedes aegypti* population at hotel compounds in Zanzibar

**Ayubo Kampango**[1,2☯]*, **Emma F. Hocke**[3,4☯], **Helle Hansson**[3,4], **Peter Furu**[5], **Khamis A. Haji**[6†], **Jean-Philippe David**[7], **Flemming Konradsen**[5], **Fatma Saleh**[8], **Christopher W. Weldon**[2], **Karin L. Schiøler**[5], **Michael Alifrangis**[3,4]

1 Sector de Estudos de Vectores, Instituto Nacional de Saúde (INS), Vila de Marracuene, Província de Maputo, Mozambique, 2 Department of Zoology and Entomology, University of Pretoria (UP), Hatfield, South Africa, 3 Center for Medical Parasitology, Department of Immunology and Microbiology, University of Copenhagen, Copenhagen, Denmark, 4 Department of Infectious Diseases, Copenhagen University Hospital (Rigshospitalet), Copenhagen, Denmark, 5 Global Health Section, Department of Public Health, University of Copenhagen, Copenhagen, Denmark, 6 Zanzibar Malaria Elimination Programme (ZAMEP), Unguja Island, Zanzibar, Tanzania, 7 Laboratoire d'Ecologie Alpine (LECA), UMR 5553, Centre National de la Recherche Scientifique (CNRS)—Université Grenoble-Alpes, Grenoble, France, 8 Department of Allied Health Sciences, School of Health and Medical Sciences, The State University of Zanzibar, Unguja Island, Zanzibar, Tanzania

☯ These authors contributed equally to this work.
† Deceased.
* akampango@gmail.com

**Data Availability Statement:** The data that support the findings of this study is provided as supplementary file S1 Data.xlsx.

## Abstract

Global efforts to control *Aedes* mosquito-transmitted pathogens still rely heavily on insecticides. However, available information on vector resistance is mainly restricted to mosquito populations located in residential and public areas, whereas commercial settings, such as hotels are overlooked. This may obscure the real magnitude of the insecticide resistance problem and lead to ineffective vector control and resistance management. We investigated the profile of insecticide susceptibility of *Aedes aegypti* mosquitoes occurring at selected hotel compounds on Zanzibar Island. At least 100 adults *Ae. aegypti* females from larvae collected at four hotel compounds were exposed to papers impregnated with discriminant concentrations of DDT (4%), permethrin (0.75%), 0.05 deltamethrin (0.05%), propoxur (0.1%) and bendiocarb (0.1%) to determine their susceptibility profile. Allele-specific qPCR and sequencing analysis were applied to determine the possible association between observed resistance and presence of single nucleotide polymorphisms (SNPs) in the voltage-gated sodium channel gene (VGSC) linked to DDT/pyrethroid cross-resistance. Additionally, we explored the possible involvement of Glutathione-S-Transferase gene (GSTe2) mutations for the observed resistance profile. In vivo resistance bioassay indicated that *Ae. aegypti* at studied sites were highly resistant to DDT, mortality rate ranged from 26.3% to 55.3% and, moderately resistant to deltamethrin with a mortality rate between 79% to and

**Funding:** This study forms a part of the EnSuZa project, Grant: 17-04-KU, funded by the Ministry of Foreign Affairs of Denmark (MFA), awarded to PF. Contributions to the study were also provided by partners in the Building Stronger Universities (BSU) programme, Phase III: 2016 – 45640, at the State University of Zanzibar (SUZA), also funded by MFA. The funders had no role in study design, data collection and analysis, decision to publish, or preparation of the manuscript.

**Competing interests:** The authors have declared that no competing interests exist. Author Khamis A. Haji was unable to confirm their authorship contributions. On their behalf, the corresponding author has reported their contributions to the best of their knowledge.

100%. However, genotyping of kdr mutations affecting the voltage-gated sodium channel only showed a low frequency of the V1016G mutation (n = 5; 0.97%). Moreover, for GSTe2, seven non-synonymous SNPs were detected (L111S, C115F, P117S, E132A, I150V, E178A and A198E) across two distinct haplotypes, but none of these were significantly associated with the observed resistance to DDT. Our findings suggest that cross-resistance to DDT/deltamethrin at hotel compounds in Zanzibar is not primarily mediated by mutations in VGSC. Moreover, the role of identified GSTe2 mutations in the resistance against DDT remains inconclusive. We encourage further studies to investigate the role of other potential insecticide resistance markers.

## Author summary

Available information on mosquito resistance to insecticides is mainly restricted to residential and public areas, whereas commercial settings, such as hotels are overlooked. This may hide the real size of an insecticide resistance problem and lead to ineffective mosquito control. We investigated insecticide susceptibility of *Aedes aegypti* mosquitoes occurring at selected hotel compounds on Zanzibar Island. We also looked at whether resistance occurred in mosquitoes with gene mutations for two proteins (voltage-gated sodium channels and glutathione-S-transferase) that are known to cause resistance to insecticides in other parts of the world. The *Ae. aegypti* mosquitoes collected from hotels were highly resistant to DDT, and moderately and possibly resistant to deltamethrin and propoxur, respectively. However, resistance to these insecticides was not linked to mutations in either of the studied genes. The presence of insecticide resistance in *Ae. aegypti* in hotel compounds on Zanzibar is concerning and shows that these areas can act as sources of resistant mosquitoes. More needs to be done to establish the underlying causes for insecticide resistance in hotel *Ae. aegypti* populations, and this information can then be used to design measures that prevent resistance from becoming more widespread on Zanzibar.

## Introduction

*Aedes*-borne arboviral diseases remain one of the most important public health threats, despite substantial investments made to reduce transmission over the last two decades [1]. *Aedes* mosquito species, *Ae. aegypti* and *Ae. albopictus*, are the most widely distributed vectors of the five most important arboviruses associated with explosive epidemics and severe infections, i.e. dengue virus (DENV), yellow fever virus (YFV), chikungunya virus (CHIKV), Zika virus (ZIKV) [2,3] and Rift Valley fever virus (RVF) [4]. An estimated 105 million dengue infections occur each year worldwide including 51 million febrile disease cases [5]. Although the global prevalence of yellow fever has been reduced by 47% since 1980 [6], new outbreaks have re-emerged in sub-Saharan Africa and South America [7–10]. A similar trend has been observed with ZIKV infections in Angola, Guinea-Bissau, Cape Verde and Ethiopia [11], and with CHIKV in several other countries within the African continent [12]. Notably, the mainland of Tanzania has experienced several DENV outbreaks within the past decade. The most recent outbreak in 2019 resulted in 6,670 confirmed cases and 13 deaths [13]. Moreover, evidence for possible active transmission of DENV in Zanzibar archipelago has increased [14–16]. This highlights the need for preparedness amid an increasing risk of *Aedes*-borne arboviral disease epidemics in Tanzania, both on the mainland and Zanzibar Islands [17,18].

General efforts to control mosquito-borne diseases (MBDs) rely heavily on chemical insecticides, mainly delivered in the form of residual spraying and long lasting insecticide treated nets [19,20]. However, large scale and intensive use of chemical insecticides, both for vector and agricultural pest control, have facilitated the emergence and expansion of resistance in vector populations against virtually all classes of insecticides registered for use in public health [21,22]. This phenomenon has threatened the gains achieved and the long-term prospect for global elimination of important MBDs, such as malaria and dengue [21]. Moreover, the higher costs associated with the development of novel, safe and efficient chemical insecticides means that existing insecticides will continue to play a pivotal role in the fight against MBDs [20]. Therefore, permanent monitoring of insecticide susceptibility profiles of local vector populations is crucial to detect the emergence of resistance, before resistant allele frequencies reach a significant level. This will support the implementation of informed and cost-saving resistance management strategies aimed at reversing resistance or maintaining susceptibility of vector species to available insecticides.

In contrast to malaria vectors [23–25], the insecticide resistance profiles of *Aedes* mosquito species occurring in sub-Saharan Africa, including Zanzibar, remains poorly documented. In addition, routine monitoring of insecticide resistance usually targets mosquitoes in and around human settlements. This may lead to the adoption of ineffective resistance management strategies that do not cover residual foci of resistant vector populations. In a recent study, Kampango et al. [26] showed that hotels on Zanzibar Island sustain abundant populations of vector mosquito species such as *Ae. aegypti* and *Ae. bromeliae.* Pyrethroids are a widely used class of insecticides, representing more than 30% of the global insecticide market [27]. They are the only class of insecticides approved for impregnation of bed nets [28] and are preferred for spraying due to their fast action, relatively low toxicity to humans and short-lived persistence in the environment [20]. In contrast, the use of DDT is restricted to indoor spraying and only recommended under particular circumstances [29]. Both pyrethroids and DDT target the transmembrane voltage-gated sodium channels (VGSC) of excitable cells leading to knockdown and death of insects on direct contact [27]. Several Single Nucleotide Polymorphisms (SNPs) in the gene coding for VGSC have been linked to pyrethroid and DDT cross resistance in different vector populations of global health concern [30,31]. Around thirteen non-synonymous SNPs in the VGSC gene, also known as knockdown resistance (*kdr*) mutations, have been discovered in pyrethroid resistant *Ae. aegypti*. However, only four of them (V410L, S989P, V1016G/I and F1534C) have been frequently associated with DDT/pyrethroid resistance [32,33]. In Africa, the SNPs resulting in F1534C and V1016I mutations have been frequently reported [34–36], whereas the V1016G mutation was only recently reported in West Africa [37]. DDT and pyrethroid cross resistance in *Ae. aegypti* have also been linked to metabolic resistance, manifested by increased over-expression of genes coding for detoxification enzymes that metabolize or sequester an insecticide before it reaches the target site [32]. This includes several genes of the cytochrome monooxygenase P450 family as well as glutathione-S-transferases (GSTs) [32]. Widespread occurrence of cross-resistance mutations can dramatically prevent or delay the implementation of control measures with devastating consequences, not least in epidemic situations. Therefore, thorough knowledge of resistance profiles, and underpinning molecular mechanisms can reduce the frequency of ineffective and wasted insecticide applications and reduce the likelihood of adverse effects on humans and the environment.

Our main objective was to determine the insecticide susceptibility of *Ae. aegypti* populations identified at selected hotel compounds on Zanzibar Island. Secondly, we investigated the potential molecular mechanisms underlying observed phenotypic resistance to pyrethroids and DDT in the local *Ae. aegypti* populations.

## Material and methods

### Ethics statement

The study received ethical clearance from the Zanzibar Health Research Institute, Ministry of Health, Zanzibar. Ref: No. ZAHREC/03/PR/Oct/2019/001. Consent was obtained from hotel management to conduct the study on their properties.

### Description of study sites

Mosquito surveys were carried out from October to November 2019 in four selected hotels located in the Southeast coastal region of Zanzibar Island (Fig 1). The study sites were previously described by Kampango et al. [26,38]. Briefly, hotels were selected according to compound size (total residential and non-residential area not less than one hectare), accessibility during low and high tourism seasons, willingness to share data and willingness to accept publication of findings. For privacy reasons, hotel names are anonymized, and hotels referred to as

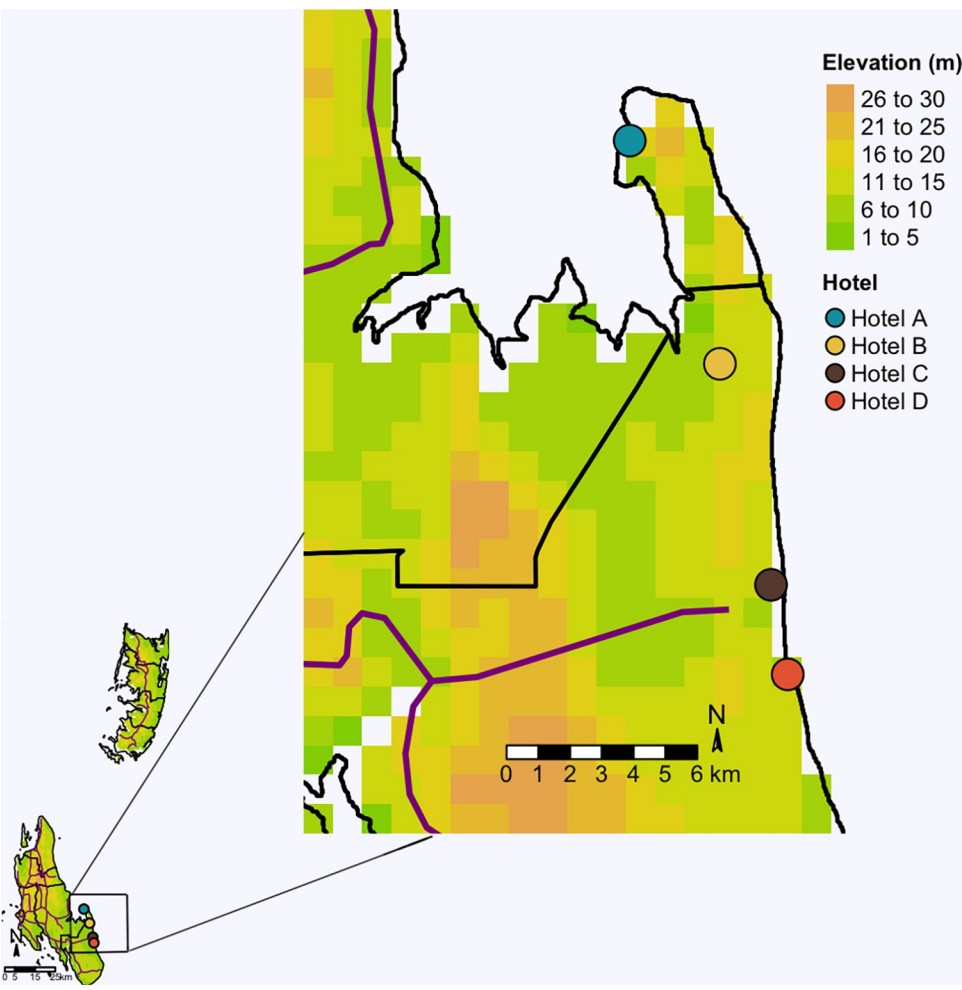

**Fig 1. Location of study sites on Zanzibar Island.** Dark magenta lines depict main roads. Zanzibar administrative borders shapefile and elevation raster were obtained freely from https://gadm.org. Shapefile of roads was freely obtained from http://www.naturalearthdata.com.

Hotel A, which occupied an area of 6.6 hectares, Hotel B (28.1 hectares), Hotel C (2.0 hectares) and Hotel D (3.6 hectares). The hotels were located at least two kilometres apart. The rainfall regime in Zanzibar archipelago is divided into two main rainy seasons; the long rainy season usually starts from mid-March to June, and the short rainy season from November to December. Average monthly precipitation ranges from 30 mm in July to 320 mm in December, and accumulated annual rainfall can reach 1600 mm [39]. *Ae. aegypti* is the most dominant mosquito species found at these hotel compounds and breeds mostly in discarded plastic containers, used tires and tree holes [26,38].

## Collection of mosquito larvae

At each hotel, immature mosquitoes were surveyed for six to seven days. Details of the adopted sampling strategy have been previous described in Kampango et al. [26,38]. For relatively small containers [water volume less than 1 litre up to 5 litres, e.g., plastic bottles and beer or soda cans (metal containers)], all larvae and pupae were sampled using pipettes. For large containers (water volume >5 litres, e.g., buckets, jerry cans and ceramic pottery) with relatively few specimens, all larvae and pupae were sampled using dippers, sweep nets or bowls. For the largest containers (water volume > 20 litres, e.g., water tanks, wells, and septic tanks), specimens were collected by 10–15 random dips/sweeps. If a container was too deep (e.g., wells) samples were collected using small, suspended buckets (approx. 5 litres), filling ten buckets from different sites of the water container.

## Sample processing and identification

Collected mosquito larvae and pupae were maintained at insectary environmental conditions of $27 \pm 2°C$ and $75 \pm 10\%$ relative humidity until they emerged as adults [40]. Larvae were fed with ground adult cat food biscuits and kept at 12h light: dark regime. Adult *Ae. aegypti* were separated from other mosquito species using distinctive morphological features according to taxonomic keys proposed by Huang [41].

## Insecticide resistance phenotypes

Insecticide susceptibility bioassays were performed according to WHO cylinder bioassay guidelines [40]. Three to five days old adult female *Ae. aegypti* were exposed to a diagnostic dose of organochlorine DDT (4%), pyrethroids permethrin (0.75%) and deltamethrin (0.05%) and carbamates propoxur (0.1%) and bendiocarb (0.1%). There is currently no consensus on insecticide diagnostic doses for *Ae. aegypti*. Therefore, we used diagnostic doses accepted for malaria vectors, as indicated by WHO insecticide bioassay guidelines [40]. The assessed insecticides were selected to represent insecticide classes in use at the hotels during the survey (S1 Table), as well as those generally used for public health, mainly deltamethrin and permethrin. DDT was intensively used for malaria control in Zanzibar in the past, but was discontinued in 1989 due to the evolution of resistance in *Anopheles* populations [42,43]. However, we decided to test DDT because the insecticide is still recommended by the WHO for vector control when resistance to other insecticides is high [29]. At least 100 female mosquitoes, divided into replicates of 20–25 specimens, were exposed to each type of insecticide per site. Simultaneously, 50 mosquitoes (25 per cylinder) exposed to papers impregnated with silicone oil acted as negative controls. The knockdown rate of mosquitoes exposed to insecticides was recorded every 10 minutes over an exposure period of one hour, after which mosquitoes were transferred into recovery cylinders and provided with cotton wool soaked in a 10% sucrose solution. The final mortality rate was estimated 24 hours after the end of exposure, and adjusted with Abbott´s correction formula when the mortality in the controls was between 5% and 20% [40]. When the mortality rate in the controls was more than

20%, the bioassay was repeated. Mosquitoes were considered susceptible to an insecticide if the final mortality rate was $\geq$ 98%, possibly resistant when the mortality rate was between 90% and 97%, and resistant if mortality was $\leq$ 90% [40]. All specimens exposed to DDT and pyrethroids (permethrin and deltamethrin) that were alive, and a random subsample of twenty specimens from all four hotels that were dead 24 hours after exposure were stored in silica gel at -20˚C for further molecular analysis.

### DNA extraction for downstream Aedes resistance genes analysis

For DNA extraction, whole mosquitoes were submerged into liquid nitrogen and ground with a pestle. The mosquito DNA was then extracted using the E-Z 96 Tissue DNA kit (Omega Bio-tek, Norcross, United States) with an adjusted initial step for arthropods as advised by the manufacturer, as follows. Initially, 180 µL of TL buffer and 20 µL of proteinase K were added and then vortexed. The samples were incubated at 56˚C overnight until the mosquito was digested (exoskeleton remains excluded). Thereafter, 200 µL of BL buffer was added, and vortexed thoroughly. Finally, 200 µL of 96% ethanol was added, and once again vortexed. After these steps, the rest of the standard extraction protocol was used according to extraction kit manufacturer instructions.

### Genotyping of polymorphisms in the VGSC gene

A subsample of 514 *Ae. aegypti* mosquitoes, comprising specimens found alive (n = 333) after being exposed to DDT (n = 296) or deltamethrin (n = 37), and dead (n = 181) after being exposed to DDT (n = 96), deltamethrin (n = 51) or permethrin (n = 34, full susceptibility), were initially analysed for the presence of *kdr* mutations by allele-specific qPCR (AS-qPCR). For the genotyping assay for polymorphisms V1016G/I, F1534C, and V410L, mosquito extracted DNA was investigated by AS-qPCR. Published primer mixes were used consisting of a common reverse primer and two specific primers targeting each polymorphic site [44–46]. The specificity of the primers was attained for the 3'-end. Additionally, a GC-tail of different length was added to the 5'-end of either the wild type or mutant primer, making them distinguishable (S2 Table). Real-time PCR reactions were carried out using PerfeCTa FastMix SYBR green (QuantaBio, Beverly, United States). The final reaction had a total volume of 20 µL consisting of 10 µL SYBR green, 2 µL primer mix of 0.2 µM (except for the V1016G mutant primer with a concentration of 0.3 µM) of each common reverse, forward mutant primer, and a forward wild-type primer, 6 µL dH$_2$O, and 2 µL DNA template. Cycling conditions for SNP F1534C were based on [47], but with a minor change in elongation temperature (from 60˚C to 72˚C), resulting in the following programme: first denaturing step of 95˚C for 10 min, then denaturing 95˚C for 15 seconds, annealing time of 54˚C for 15 seconds, and elongation time of 72˚C for 30 seconds for 40 cycles [47]. Cycling conditions for the SNP causing the V1016I change consisted of a denaturing step of 95˚C for 10 min, then denaturing 94˚C for 30 seconds, annealing time of 62˚C for 1 min, and elongation time of 72˚C for 45 seconds for 40 cycles [44]. Cycling conditions for V1016G consisted of a first denaturing step of 94˚C for 10 min, then denaturing 94˚C for 30 seconds, annealing time of 55˚C for 30 sec, and elongation time of 72˚C for 30 seconds for 40 cycles [45]. Cycling conditions for V410L consisted of a first denaturing step of 95˚C for 10 min, then denaturing 95˚C for 1 min, annealing time of 56˚C for 20 sec, and elongation time of 72˚C for 20 seconds for 40 cycles [46].

### Sequencing of domains in the VGSC gene

Genetic diversity in partial sections of segment S6 in domains I-IV of the VGSC gene were investigated by Sanger sequencing. A total subset of 85 mosquitoes was used to partially

sequence segment 6 of domain I-IV of the VGSC to verify the qPCR results and to explore the VGSC gene for other known (S989P, A1007G, I1011M/V, and T1520I) and possibly unknown SNPs. The subset of mosquitoes included 20 sequenced samples in domain I (10 dead, 10 alive), 85 sequenced samples in domain II (25 dead, 60 alive), 28 sequenced samples in domain III (12 dead, 16 alive), and 19 sequenced in domain IV (10 dead, 9 alive). The sections were amplified by conventional PCR, in a reaction volume of 20 μL consisting of 11 μL dH$_2$O, 1 μL of each forward and reverse primer with a concentration of 0.20 μM, 5 μL master mix, and 2 μL of DNA. Each domain was amplified with domain-specific primers (S3 Table). Briefly, for domain I: initial denaturation of 95˚C for 10 min, followed by 35 cycles of denaturation at 94˚C for 30 sec, annealing at 60˚C for 30 sec, and elongation at 72˚C for 1 min with a final elongation of 5 min. For domains II, III, and IV: initial denaturation at 95˚C for 10 min, followed by denaturation at 95˚C for 30 sec, annealing at 55˚C for 1 min and elongation at 72˚C for 1 min repeated for 40 cycles followed by a final elongation for 1 min and 30 seconds. Amplicons were verified by electrophoresis (1% agarose gel) and purified using MicroElute Cycle-Pure Kit (Omega Bio-Tek, Norcross, United States) (S3 Table). Concentrations in ng/μL of purified amplicons were measured three times for better precision on the NanoDrop 2000/ 2000c Spectrophotometer (Thermo Fisher Scientific, Waltham, United States), adjusted prior to commercial sequencing (Eurofins, Luxembourg). All sequences were compared to reference sequences in Genbank using the MEGA-BLAST algorithm [48]. Consensus sequences were constructed using BioEdit 7.2 [49] with a minimum of 20 bp overlap and 85% similarity. All sequences were then aligned by the MUSCLE algorithm [50] using the Molecular Evolutionary Genetics Analysis version 10 (MEGA X) [51].

## DNA sequencing of the Ae. aegypti GSTe2 gene

For amplification and sequencing of the GSTe2 gene, specific primers were constructed according to the reference gene LOC110676856 in the *Ae. aegypti* AaegL5.0 assembly (https:// www.ncbi.nlm.nih.gov/genome/gdv/browser/genome/?id=GCF_002204515.2). Two primers were constructed to amplify the whole gene sequence and an additional four primers were constructed for Sanger sequencing of the whole gene (S1 Fig and S3 Table). The PCR reaction was carried out with a total volume of 50 μL consisting of 2 μL of each amplification primer (0.20 μM) (S2 Table), 17 μL of TEMPase Hot Start DNA Polymerase (VWR, Radnor, United States), 24 μL dH$_2$O, and 5μL of DNA. The amplification included initial denaturation at 94˚C for 15 minutes, cycling consisted of denaturation at 94˚C for 1 min, annealing at 55˚C for 1 min, elongation at 72˚C for 2 min for 35 cycles followed by a final elongation at 72˚C for 10 min. Amplicons were verified by electrophoreses (1% agarose gel) and purified using MicroElute Cycle-Pure Kit (Omega Bio-Tek, Norcross, United States). The concentration of purified amplicons was measured three times on a NanoDrop 2000/2000c Spectrophotometer (Thermo Fisher Scientific, Waltham, United States), and the average concentration used for further calculations. Concentrations in ng/μL of purified amplicons were measured on the NanoDrop 2000/2000c Spectrophotometer (Thermo Fisher Scientific, Waltham, United States), adjusted prior to commercial sequencing (Eurofins, Luxembourg). The procedure for analysis of GSTe2 sequences was similar to analysis of those VGSC as mentioned above.

## Quantification of GSTe2 gene variants

For detection and quantification of copy number variations (CNV) of the GSTe2 gene, the concentration of 5 μL of dsDNA was measured on a Qubit 4 fluorometer (Thermo Fischer Scientific, Waltham, United States) using the Qubit dsDNA HS Assay Kit (Invitrogen, Waltham, United States). Since no CNV controls were available, the concentration of the samples was

equalized to compare differences in signal from resistant and susceptible mosquito samples. The concentration of the dsDNA was adjusted to be 3 ng/μL. A CNV specific qPCR for GSTe2 was then performed using specific primers (S2 Table) at a concentration of 0.32 μM. The qPCR assay used a final volume of 20 μL consisting of 10 μL PerfeCTa FastMix SYBR green (QuantaBio, Beverly, United States), 6 μL of $dH_2O$, 1μL of each primer, and 2μL of adjusted DNA. The following qPCR conditions were used: an initial denaturation at 95˚C for 10 minutes, cycling consisted of 95˚C denaturation for 15 seconds, annealing at 55˚C for 30 seconds repeated for 40 cycles.

## Statistical analysis

Log-probit regression analyses were applied to predict the probability of *Ae. aegypti* being knocked down by each insecticide during exposure, and to estimate the average time necessary to knockdown 50% ($KDT_{50}$) and 95% ($KDT_{95}$) of *Ae. aegypti* populations when in contact with each type of insecticide. Regression models were fitted using the R software package drc v. 3.0–1 [52]. Fisher exact tests were applied to determine whether the difference between the frequencies of insecticide resistance polymorphisms was significant. A simple t-test was applied for CNV to investigate any significant difference in qPCR Ct-values in the GSTe2 gene between susceptible and resistant *Ae. aegypti* samples.

## Results

### Insecticide susceptibility

A total of 2,561 *Ae. aegypti* females were tested (Hotel A: n = 400 specimens; Hotel B: n = 836; Hotel C: n = 775; Hotel D: n = 550). The results of probit regression analysis are depicted in Fig 2 and summarized in Table 1. Mosquitoes from Hotel A were tested with only four insecticides due to too few specimens being available. In general, *Ae. aegypti* specimens from all four hotels showed the slowest knockdown rate when exposed to DDT (Fig 2 and Table 1). The estimated average knockdown time $KDT_{50}$ of DDT exposed mosquitoes varied from 61.1 (95% CI, 56.6–65.5) among specimens from Hotel D, to a maximum of 152.7 (95% CI, 84.7–220.7) minutes for specimens from Hotel B. Mortality rates estimated twenty-four hours post-exposure ranged from 26.3% (Hotel B) to 55.3% (Hotel D) indicating presence of resistance to DDT in *Ae. aegypti* populations from all four hotels (Table 1). Results also suggest moderate resistance to deltamethrin, with mortality rates ranging from 79% (Hotel B) to 100% (Hotel D), and possible low levels of resistance to propoxur in the population from Hotel D, where the mortality rate was 96%. Full susceptibility was observed with permethrin and bendiocarb across all study sites (Table 1).

### Frequency of SNPs in the VGSC gene

Molecular analysis targeting mutations in the VGSC identified only five mosquitoes that were heterozygous at V1016G. These five mosquitoes were all DDT resistant as determined by the insecticide susceptibility bioassay, and all were from the same site (Hotel B). Results of nucleotide sequences of partially sequenced segment 6 of domain I-IV of the VGSC were identical to the results obtained by qPCR. No previously described or new SNPs were detected in the VGSC gene (Fig 3).

### Polymorphisms and CNV in the GSTe2 gene

A total of 57 mosquitoes were PCR positive for presence of the GSTe2 gene. 47 confirmed positive samples were chosen for purification and sequencing, and 21 mosquitoes were

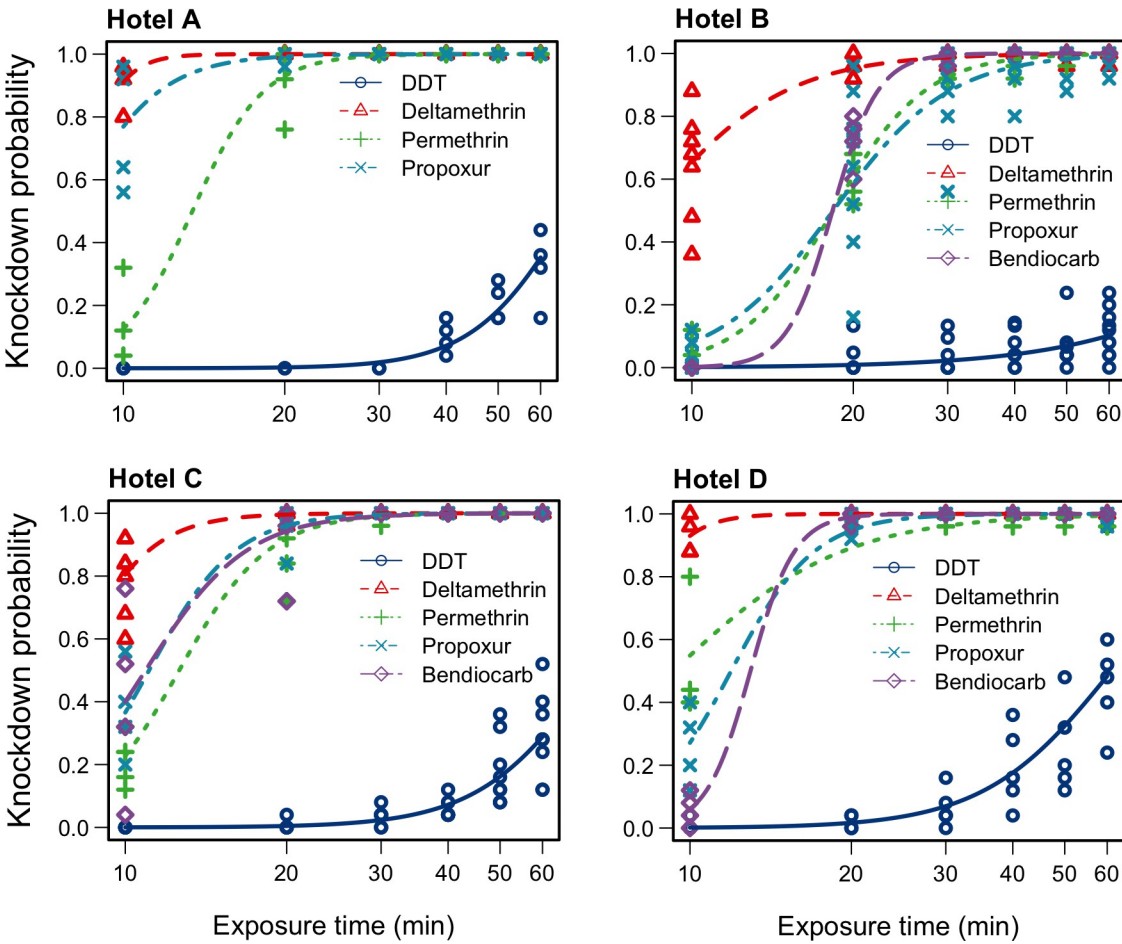

**Fig 2. Knockdown probability of *Aedes aegypti* exposed to discriminant doses of insecticides.** Consistent delayed knockdown of *Ae. aegypti* exposed to DDT is visible at all sites, which indicates resistance.

successfully sequenced in the whole GSTe2 gene (Five susceptible and 16 resistant). Seven non-synonymous SNPs resulting in the following changes: L111S, C115F, P117S, E132A, I150V, E178A and A198E. These substitutions were consistently distributed across two distinct haplotypes. Thirteen of the 21 sequenced haplotypes belonged to haplotype 1 (4 susceptible and 7 resistant) and 8 belonged to haplotype 2 (1 susceptible and 7 resistant) (S2 Fig). Haplotype 2 showed low similarity (~90% homology) to available sequences in Genbank, whereas haplotype 1 showed ~96% homology. However, no statistically significant association between haplotype variant and insecticide resistance was found (p = 0.65). Regarding CNV, no statistically significant difference was found between the qPCR Ct values measured for 14 DDT susceptible (mean Ct value = 20.70 (range 19.08–22.34) and 15 DDT resistant (mean Ct value = 20.56 (range 19.43–21.76) *Ae. aegypti* samples (p = 0.61).

## Discussion

Accurate knowledge of mosquito susceptibility to available insecticidal control measures and the underlying environmental and biological mechanisms conferring resistance is crucial for selection of appropriate vector control strategies and implementation of cost-saving resistance management practices. To the best of our knowledge, this is the first attempt to investigate the

**Table 1. Estimates of mortality rates and average exposure time necessary to knockdown Aedes aegypti exposed to insecticides.** Shaded rows indicate resistance or suspected resistance requiring confirmation.

| Hotel | Insecticide | Total tested/ Replicate No. | $KDT_{50}$ (± 95 % CI) | $KDT_{95}$ (± 95 % CI) | Number dead | Mortality rate |
|---|---|---|---|---|---|---|
| Hotel A | DDT | 100/4 | 68.1 (61 - 75.1) | 125.3 (90.6 - 160.1) | 39 | 39% |
| | Deltamethrin | 100/4 | 8.1 (1.6 - 14.6) | 10.6 (8.2 - 13) | 99 | 99% |
| | Permethrin | 100/4 | 7.95 (6.66 - 9.24) | 13.94 (11.51 - 16.38) | 100 | 100% |
| | Propoxur | 100/4 | 7.9 (6.7 - 9.2) | 13.9 (11.5 - 16.4) | 100 | 100% |
| Hotel B | DDT | 236/10 | 152.7 (84.7 - 220.7) | 541 (53.7 - 1028.3) | 62 | 26% |
| | Deltamethrin | 200/8 | 8.3 (7.4 - 9.2) | 19.7 (17.4 - 22.1) | 158 | 79% |
| | Permethrin | 200/8 | 18.4 (17.4 - 19.3) | 38.3 (35.2 - 41.3) | 196 | 98% |
| | Propoxur | 100/4 | 18.3 (17.1 - 19.6) | 32.7 (29.4 - 36) | 99 | 99% |
| | Bendiocarb | 100/4 | 18.3 (17.3 - 19.3) | 24.3 (22.2 - 26.5) | 98 | 98% |
| Hotel C | DDT | 225/10 | 74.9 (67.4 - 82.3) | 154.3 (115.2 - 193.3) | 96 | 43% |
| | Deltamethrin | 200/8 | 7.9 (6.8 - 8.9) | 13 (11.3 - 14.6) | 180 | 90% |
| | Permethrin | 150/6 | 12.6 (11.9 - 13.4) | 22 (19.8 - 24.3) | 150 | 100% |
| | Propoxur | 100/4 | 10.9 (10.1 - 11.8) | 20.6 (17.6 - 23.6) | 100 | 100% |
| | Bendiocarb | 100/4 | 11.1 (10.3 - 11.8) | 19.1 (16.4 - 21.9) | 100 | 100% |
| Hotel D | DDT | 150/6 | 61.1 (56.6 - 65.5) | 136.8 (106.8 - 166.9) | 83 | 55% |
| | Deltamethrin | 100/4 | 7.4 (2.4 - 12.4) | 10.4 (9.1 - 11.7) | 100 | 100% |
| | Permethrin | 100/4 | 11.9 (11.1 - 12.8) | 20.2 (17.5 - 22.9) | 100 | 100% |
| | Propoxur | 100/4 | 9.3 (7.9 - 10.7) | 27.1 (22.4 - 31.8) | 96 | 96% |
| | Bendiocarb | 100/4 | 13 (11.9 - 14.1) | 17.1 (14.7 - 19.6) | 100 | 100% |

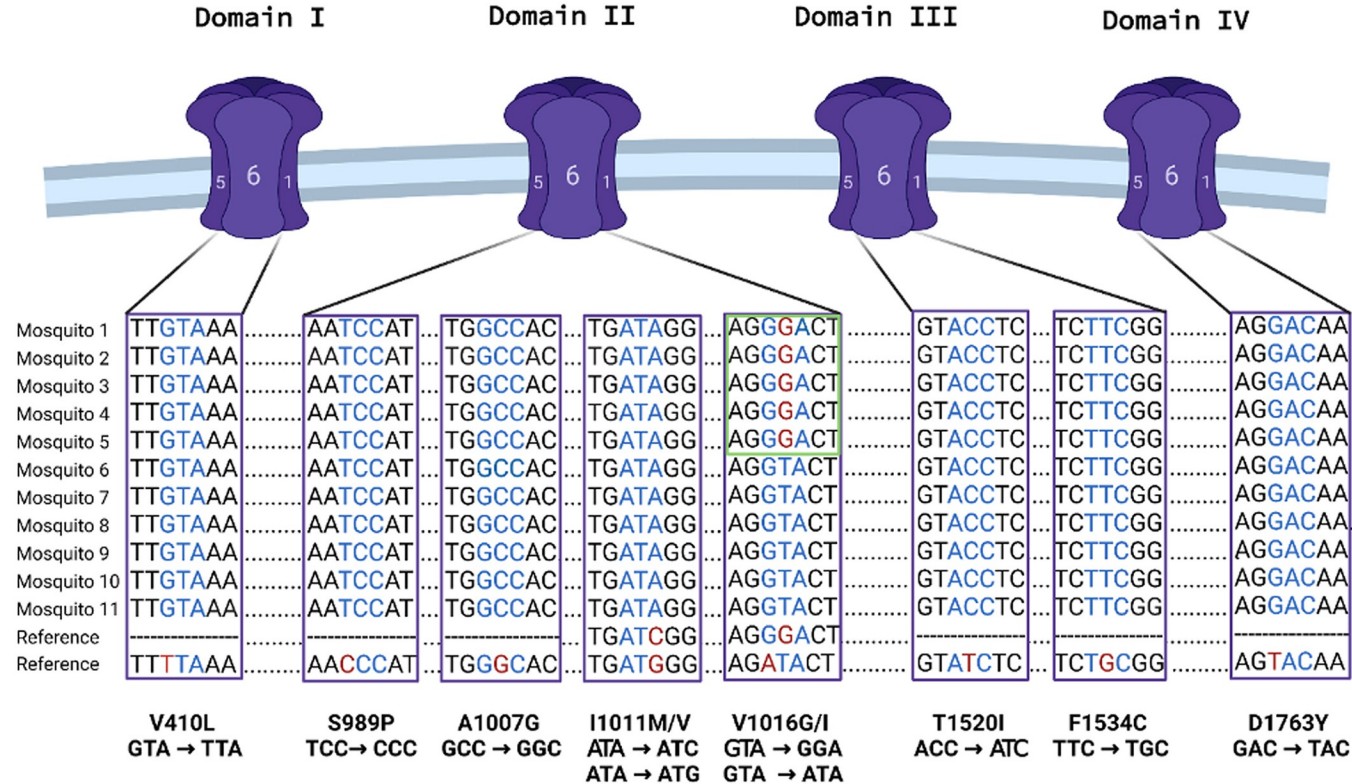

**Fig 3. Alignment of eleven representative nucleotide sequences from DDT and deltamethrin resistant *Ae. aegypti* samples.** The reference sequences are constructed to contain mutant codons and the sequence obtained from each of the VGSC structure domains (I—IV) are also shown. The green box highlights the five V1016G mutants detected through allele-specific qPCR in segment six (S6) of domain II that were verified by Sanger sequencing.

status of insecticide resistance and possible association to molecular alterations in *Ae. aegypti* populations of the Zanzibar archipelago. *In vivo* susceptibility bioassays indicated that *Ae. aegypti* populations at four hotel sites were highly resistant to DDT, moderately resistant to deltamethrin and possibly resistant to propoxur, while fully susceptible to permethrin and bendiocarb. The insecticide susceptibility status of arboviral vectors, not least *Ae. aegypti*, remains poorly investigated in Tanzania, as indeed is most of the East African region (https://aedes.irmapper.com), with the few published reports limited to phenotypic characterisation of resistance. This includes a recent survey of *Ae. aegypti* in the Ifakara area of south-eastern Tanzania, showing full susceptibility to deltamethrin, permethrin, bendiocarb and pirimiphos-methyl [53] and a study from Dar es Salaam, in which *Aedes aegypti* displayed resistance to deltamethrin and lambda-cyhalothrin, but varying susceptibility to permethrin [54].

The lowest mortality rate of mosquitoes exposed to DDT was observed at Hotel B, whereas mortality rates from the remaining three hotels were statistically comparable. DDT was for many years an integral part of anti-malaria house-spraying campaigns in Zanzibar. However, it was fully withdrawn in 1988 due to high levels of DDT resistance in *An. gambiae s.l.*, the dominant malaria vector complex in the region [42]. An exploratory survey of mosquito control practices at hotels, carried out prior to this study, did not find any evidence of recent use of DDT or analogue insecticides at the study hotels (S1 Table). However, two hotels (Hotel A and Hotel B) had implemented periodic outdoor spraying with an emulsified concentrate of the organophosphate dichlorvos and the pyrethroids deltamethrin, D-trans-allethrin (S1 Table). This may somehow explain the high level of deltamethrin resistance observed at hotel B, in particular. Space spraying with deltamethrin was also performed periodically at Hotel B. The four hotels performed aerosol (indoor) spraying using locally available commercial canned mixtures of the pyrethroids imiprothrin and cypermethrin, or with a formulation of the carbamate, propoxur. All of the above-mentioned insecticides are currently registered for use in Tanzania [55]. With 40 years having elapsed since the last widespread use of DDT to control *Anopheles* mosquitoes it is notable that *Ae. aegypti* in Zanzibar are resistant to this insecticide. However, mosquitoes can retain resistance to DDT over long periods. For instance, field studies in India on the malaria vectors *An. stephensi* and *An. culicifacies* showed that these species had retained resistance to DDT thirty years after the use of this insecticide was officially terminated [56]. Contrarily, other published evidence shows that slow reversion toward DDT susceptibility can occur in wild populations of *An. stephensi* and *Ae. aegypti* when the DDT pressure is removed [57,58]. As such, we propose that the past intense use of dieldrin (1958–1961), and thereafter DDT (until 1988), as part of malaria eradication campaigns in Zanzibar [42,59], may have led to unintended selection of DDT resistance in local *Ae. aegypti* populations.

Allele-specific qPCR-based assays detected the presence of heterozygous V1016G mutant alleles in the VGSC gene of five specimens of *Ae. aegypti* (found at Hotel B). No other investigated point mutations commonly found in *Ae. aegypti* were detected. This finding contrasts with the outcome of our phenotypic resistance bioassay, which indicated higher resistance to DDT and moderate resistance to deltamethrin. The geographical occurrence of the V1016G mutation is expanding in the Middle East [60], southeast Asia [45,61–64] and Latin America [65]. Notably, it was recently reported in Cameroon, West Africa [37]. The origin of V1016G/I mutations in African populations of *Ae. aegypti* remains poorly understood. Fan et al. [66] proposed that the variant may have arisen from genetic recombination in the evolution of *kdr* alleles or by introduction from another continent. We did not observe a clear association between the presence of the V1016G alleles and reduced susceptibility to DDT and deltamethrin in *Ae. aegypti* from Zanzibar. The lower frequency of V1016G alleles suggests recent introduction or occurrence of this type of *kdr* mutation in *Ae. aegypti* populations at the

studied sites. It may also suggest reduced fitness of V1016G alleles and might be unfavourable to maintain. As noted earlier, there is intensive pesticide use at the studied hotels, which may have applied selection pressure for resistance in the *Ae. aegypti* populations. Therefore, further investigations are encouraged to understand factors driving the emergence and propagation of *kdr* mutations in order to prevent their successful establishment and expansion throughout the Zanzibar archipelago.

Several genes of the family cytochrome P450 and glutathione-S-transferases (GSTes) have been associated with pyrethroid and DDT resistance in *Ae. aegypti* populations [35,67,68]. For instance, a study by Helvecio et al. [69] found that several SNPs in GSTe2 correlated with temephos resistance in *Ae. aegypti*. However, in this study, sequencing analyses indicated no significant correlation between *Ae. aegypti* resistance to DDT and presence of GSTe2 SNPs observed (L111S, C115F, P117S, E132A, I150V, E178A, and A198E). In Senegal, Sene et al. [68] also reported higher phenotypic resistance to DDT and permethrin in *Ae. aegypti* but, similar to this study, no *kdr* mutations were detected. The authors found overexpression of genes of the P450 family namely *CYP6BB2*, *CYP9J26* and *CYP9J32*, associated with resistance to pyrethroids and DDT, and also high overexpression of the GSTD4 glutathione-S-transferase gene, associated with resistance to DDT. This study did not investigate the potential role of P450 family genes since, in general, *Ae. aegypti* showed susceptibility to the two types of pyrethroids tested, excepting at Hotel B where resistance to deltamethrin was detected. Using lab-based resistance selection experiments, Smith et al. [67] showed that cytochrome P450 genes can also mediate cross resistance to pyrethroids and organophosphates in *Ae. aegypti* either in the presence or absence of *kdr* mutations, underscoring the complexity of molecular mechanisms of insecticide resistance in *Ae. aegypti* populations. No significant association was detected between GSTe2 polymorphisms and phenotypic resistance to DDT, suggesting that a potentially different resistance mechanism may be involved. There are many other poorly investigated resistance mechanisms to pyrethroids and DDT, such as the recently described cuticular-based resistance to pyrethroids and DDT in *Ae. aegypti* from Cameroon [70]. Finally, previous studies with *An. arabiensis* and *An. gambiae s.s.* reported an eight-fold higher increase of DDT-dehydrochlorinase activity of glutathione S-transferases in resistant populations compared to susceptible ones [71]. As such, we encourage further investigations into the probable involvement of other, less common, mechanisms that might underpin pyrethroid/DDT resistance in *Ae. aegypti* and similar vectors of public health concern.

## Concluding remarks

Our study shows that resistance to pyrethroids and DDT is present in *Ae. aegypti* of Zanzibar. It also shows, for the first time, the presence of the *kdr* mutation V1016G in the region. However, any association between the V1016G mutation and resistance to pyrethroids and DDT in *Ae. aegypti* populations remains inconclusive due to low detection frequency of the resistant allele. Emergence of V1016G *kdr* mutations in Zanzibar should be of great concern given the intense use of pyrethroids in and around hotels. Therefore, efforts should be made to reduce selection pressure by replacing the use of chemical control by environmental mosquito management practices. Successful expansion of V1016G throughout the archipelago would compromise the effectiveness of vector control by both type 1 and type 2 pyrethroids with devastating consequences in the event of an *Aedes*-borne disease epidemic. Our findings also highlight the need for continued and comprehensive investigations on other potential mechanisms for pyrethroid resistance in local *Ae. aegypti* populations. Moreover, the risk of large-

scale use of insecticides for mosquito control at private facilities, outside direct government control, should be thoroughly investigated.

## Supporting information

**S1 Fig. Overview of the GSTe2 transcript and constructed primers used for amplification and sequencing.** Arrows indicate direction of synthesis.
(TIF)

**S2 Fig. Alignment starting from position 110 to 200 of two representative haplotype protein sequences with SNPs L111S, C115F, P117S, E132A, I150V, E178A, A198E present.**
(TIFF)

**S1 Table. Mosquito control practices, type, quantity and frequency of chemical insecticides utilization at hotel compounds.**
(DOCX)

**S2 Table.** Primers used for allele-specific qPCR of the VGSC (A) and CNV assay of the GSTe2 gene (B).
(DOCX)

**S3 Table.** Primers used to amplify and sequence partial segment 6 in domain I-IV (A) and whole GSTe2 gene (B).
(DOCX)

**S1 Data. Excel format dataset containing records of *in vivo* susceptibility bioassays.**
(XLSX)

## Acknowledgments

The authors would like to thank all members of the EnSuZa project for constructive discussions and support during the study. We would also like to thank the hotel managers and the Zanzibar Association of Tourism Investors for their unconditional support and permanent collaboration.

## Author Contributions

**Conceptualization:** Ayubo Kampango, Peter Furu, Christopher W. Weldon, Karin L. Schiøler, Michael Alifrangis.

**Data curation:** Ayubo Kampango, Emma F. Hocke, Helle Hansson.

**Formal analysis:** Ayubo Kampango, Emma F. Hocke.

**Funding acquisition:** Peter Furu.

**Investigation:** Ayubo Kampango, Emma F. Hocke, Helle Hansson, Khamis A. Haji, Jean-Philippe David, Flemming Konradsen, Fatma Saleh.

**Methodology:** Ayubo Kampango, Emma F. Hocke, Jean-Philippe David, Flemming Konradsen, Fatma Saleh, Christopher W. Weldon, Karin L. Schiøler, Michael Alifrangis.

**Project administration:** Peter Furu, Karin L. Schiøler, Michael Alifrangis.

**Resources:** Flemming Konradsen, Fatma Saleh, Christopher W. Weldon, Karin L. Schiøler, Michael Alifrangis.

**Software:** Ayubo Kampango, Emma F. Hocke.

**Supervision:** Ayubo Kampango, Peter Furu, Khamis A. Haji, Fatma Saleh, Christopher W. Weldon, Karin L. Schiøler, Michael Alifrangis.

**Validation:** Ayubo Kampango, Emma F. Hocke, Helle Hansson.

**Visualization:** Ayubo Kampango, Emma F. Hocke, Helle Hansson.

**Writing – original draft:** Ayubo Kampango.

**Writing – review & editing:** Ayubo Kampango, Emma F. Hocke, Helle Hansson, Peter Furu, Khamis A. Haji, Jean-Philippe David, Flemming Konradsen, Fatma Saleh, Christopher W. Weldon, Karin L. Schiøler, Michael Alifrangis.

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
