## [Decision Letter · Decision Letter 0]

7 Mar 2022

Dear Mr Kampango,

Thank you very much for submitting your manuscript "High DDT resistance contrasts with low frequency of kdr mutations and no evidence of Glutathione-S-transferase (GST) gene mutations in Aedes aegypti populations at hotel compounds in Zanzibar" for consideration at PLOS Neglected Tropical Diseases. As with all papers reviewed by the journal, your manuscript was reviewed by members of the editorial board and by several independent reviewers. The reviewers appreciated the attention to an important topic. Based on the reviews, we are likely to accept this manuscript for publication, providing that you modify the manuscript according to the review recommendations. 

Sincerely,

Mariangela Bonizzoni

Associate Editor

Tereza Magalhaes

Deputy Editor

Reviewer's Responses to Questions

**Key Review Criteria Required for Acceptance?**

**Methods**

-Are the objectives of the study clearly articulated with a clear testable hypothesis stated?

-Is the study design appropriate to address the stated objectives?

-Is the population clearly described and appropriate for the hypothesis being tested?

-Is the sample size sufficient to ensure adequate power to address the hypothesis being tested?

-Were correct statistical analysis used to support conclusions?

-Are there concerns about ethical or regulatory requirements being met?

Reviewer #1: The objectives are clear and the methodology appropriate, with sufficient numbers and statistics

Reviewer #2: (No Response)

**Results**

-Does the analysis presented match the analysis plan?

-Are the results clearly and completely presented?

-Are the figures (Tables, Images) of sufficient quality for clarity?

Reviewer #1: the analysis matches the analysis plan

Reviewer #2: (No Response)

**Conclusions**

-Are the conclusions supported by the data presented?

-Are the limitations of analysis clearly described?

-Do the authors discuss how these data can be helpful to advance our understanding of the topic under study?

-Is public health relevance addressed?

Reviewer #1: the conclusions are supported by the data that are poresented

Reviewer #2: (No Response)

**Editorial and Data Presentation Modifications?**

Reviewer #1: (No Response)

Reviewer #2: (No Response)

**Summary and General Comments**

Reviewer #1: It is a good manuscript, well written and presented data and technically well executed study. Reporting some interesting information about the lack of tight association of known DDT resistance markers with resistance phenotypes. Perhaps some basic synergism data, in case still possible to include, would have provided some more insights into the possible mechanisms (detoxification vs target site resistance)

Reviewer #2: (No Response)

PLOS authors have the option to publish the peer review history of their article (what does this mean?). If published, this will include your full peer review and any attached files.

Reviewer #1: No

Reviewer #2: Yes: Ashwaq Madani Alnazawi

Figure Files:

Data Requirements:

Reproducibility:

References

---

## [Editor Report · Decision Letter 1]

25 Mar 2022

Dear Mr Kampango,

We are pleased to inform you that your manuscript 'High DDT resistance without apparent association to kdr and Glutathione-S-transferase (GST) gene mutations in Aedes aegypti population at hotel compounds in Zanzibar' has been provisionally accepted for publication in PLOS Neglected Tropical Diseases.

Best regards,

Mariangela Bonizzoni

Associate Editor

Tereza Magalhaes

Deputy Editor

---

## [Editor Report · Acceptance letter]

25 Apr 2022

Dear Mr Kampango,

We are delighted to inform you that your manuscript, "High DDT resistance without apparent association to kdr and Glutathione-S-transferase (GST) gene mutations in Aedes aegypti population at hotel compounds in Zanzibar," has been formally accepted for publication in PLOS Neglected Tropical Diseases.

Best regards,

Shaden Kamhawi

co-Editor-in-Chief

Paul Brindley

co-Editor-in-Chief
